# Development of Mucosal PNAd^+^ and MAdCAM-1^+^ Venules during Disease Course in Ulcerative Colitis

**DOI:** 10.3390/cells9040891

**Published:** 2020-04-06

**Authors:** Britt Roosenboom, Ellen G. van Lochem, Jos Meijer, Carolijn Smids, Stefan Nierkens, Eelco C. Brand, Liselot W. van Erp, Larissa G.J.M. Kemperman, Marcel J.M. Groenen, Carmen S. Horjus Talabur Horje, Peter J. Wahab

**Affiliations:** 1Crohn & Colitis Centre Rijnstate, Department of Gastroenterology and Hepatology, Rijnstate Hospital, 6815 AD Arnhem, The Netherlands; 2Department of Microbiology and Immunology, Rijnstate Hospital, 6815 AD Arnhem, The Netherlands; 3Department of Pathology, Rijnstate Hospital, 6815 AD Arnhem, The Netherlands; 4U-DAIR and Center for Translational Immunology, University Medical Center Utrecht, Utrecht, The Netherlands; 5Department of Gastroenterology and Hepatology and Center for Translational Immunology, University Medical Center Utrecht, 3584 CX, Utrecht, The Netherlands

**Keywords:** TLOs, HEVs, IBD

## Abstract

PNAd and MAdCAM-1 addressins on venules are of importance in T-cell homing and potential therapeutic targets in ulcerative colitis (UC). Normally, PNAd^+^ high endothelial venules (HEVs) are only present in lymphoid organs, whereas small numbers of MAdCAM-1^+^ venules can be seen in non-lymphoid tissue. We aimed to study their presence in the intestinal mucosa of UC patients at diagnosis and during follow-up, and their correlation with disease activity. Colonic biopsy specimens of 378 UC patients were analyzed by immunohistochemistry for CD3, CD20, ERG, MECA-79 (PNAd) and MECA-376 (MAdCAM-1) and compared to healthy controls (HC). The proportion of PNAd^+^HEVs in UC at diagnosis was 4.9% (IQR 2.0%–8.3%), while none were detected in HC. During follow-up, PNAd^+^HEVs completely disappeared in remission (*n* = 93), whereas the proportion in active disease was similar to baseline (*n* = 285, *p* = 0.39). The proportion of MAdCAM-1^+^venules in UC at baseline was 5.8% (IQR 2.6–10.0). During follow-up, the proportion in remission was comparable to diagnosis, but upregulated (7.5% (IQR 4.4–10.9), *p* = 0.001) in active disease. In conclusion, PNAd^+^HEVs appear in UC during active inflammation which could thus serve as a marker for disease activity, whereas MAdCAM-1^+^venules remain present after inflammation is resolved and increase after subsequent flares, reflecting chronicity and potentially serving as a therapeutic target.

## 1. Introduction

Ulcerative colitis (UC) is known to have a heterogenic phenotype reflected by differences in disease location and severity, age of disease onset and response to treatment [1]. There are several therapeutic agents available to reduce symptoms or to prevent progression of disease in patients with UC. However, the response to treatment differs, suggesting that distinct inflammatory mechanisms drive the course of the disease [2,3,4].

In healthy gut mucosa, naive- (T_n_) and central memory T cells (T_cm_) migrate to secondary lymphoid organs (SLOs) by tethering and rolling on specialized cuboidal formed high endothelial venules (HEVs) [5,6]. This process is facilitated through the binding of L-selectin on the surface of T cells to peripheral node addressin (PNAd) on HEVs [7]. Within SLOs, T cells become activated effector cells (T_em_) and migrate through blood vessels to their site of action, such as the gut mucosa. The adhesion molecule integrin alpha4beta7 (α4β7) on T_em_ cells plays a crucial role in controlling this migration process to the intestine by binding to mucosal vascular addressin cell adhesion molecule-1 (MAdCAM-1), a 60-kD glycoprotein, which is expressed on venules in Peyer’s patches, mesenteric lymph nodes and on flattened venular endothelial cells in the intestinal lamina propria [8]. MAdCAM-1 contributes to lymphocyte homing by serving as a cell adhesion molecule, not only by binding α4β7^+^, but to a lesser extent also by binding L-selectin^+^ and/or α4β1^+^ lymphocytes to the luminal surface of venules, and as a vascular addressin for the tethering and rolling of lymphocytes [9].

In contrast to the relative absence of T cells in non-inflamed gut mucosa, they are found in high numbers in the inflamed gut of UC patients reflecting the diffuse chronic inflammatory cell infiltrate [10]. A possible critical step needed to generate this infiltrate, is the morphological and functional change of postcapillary venules into HEVs in non-lymphoid tissue. HEVs are proposed to be absent in the non-lymphoid tissue of healthy gut mucosa. Therefore, their presence might serve as a marker of newly formed tertiary lymphoid organs (TLO), with a quite similar histological appearance to SLOs [11]. These newly developed lymphoid organs might facilitate the homing and reactivation of T cells independent of SLOs in chronic inflamed mucosa [12].

Currently, little is known about the presence of PNAd^+^ and MAdCAM-1^+^ venules in the colon of UC patients and their role in the pathogenesis and disease course of UC [13]. During active disease in UC patients, the induction of colonic PNAd^+^ HEVs was associated with a greater influx of T_n_ and T_cm_ cells and correlated with the intensity of inflammation based on Ulcerative Colitis Disease Activity Index (UCDAI) scores in a small group of patients [14,15]. In another small cohort of patients, MAdCAM-1^+^venules were suggested to be upregulated in active UC compared to HC, with no differences in numbers of MAdCAM-1^+^ venules between patients with active disease and remission [16]. These adhesion molecules and vascular addressins are attractive targets in the treatment of UC since they specifically facilitate the migration of lymphocytes to the gut mucosa, which plays a vital role in the pathogenesis of UC [17]. Anti-α4β7 integrin (Vedolizumab) is an effective therapy to induce and maintain clinical and endoscopic remission [18]. In addition, the effect of antibodies against MAdCAM-1 (Ontamalimab), which prevent the migration of T cells to the gut by blocking the same ‘homing pathway’ as anti-α4β7 integrin antibodies, has been studied in a phase II trial with promising results [19]. However, not all patients respond to treatments interfering with the homing of T cells by blocking α4β7 and MAdCAM-1 [20,21,22], probably because of the simultaneous presence of PNAd^+^HEVs, serving as an entrance for colitogenic T_n_ cells [23]. In contrast to anti-MAdCAM-1 therapy, anti-PNAd therapies have not yet been investigated in humans.

In the present study, we evaluated the presence of both PNAd on HEVs and MAdCAM-1 on HEVs and flattened venules in the gut mucosa of UC patients at initial diagnosis and during follow-up endoscopy. We aimed to investigate the correlation between the expression of PNAd and MAdCAM-1 on colonic mucosal venules with histologic disease activity at diagnosis and during disease flares and their potential to predict the course of disease in patients with UC.

## 2. Patients and Methods

### 2.1. Study Population

Newly diagnosed untreated UC patients between 2000 and 2018 at the Department of Gastroenterology and Hepatology in Rijnstate Hospital in Arnhem, a secondary care center in the Netherlands, were retrospectively included in the study. In patients with symptoms including rectal blood loss, chronic diarrhea, abdominal pain or weight loss, the diagnosis of UC was confirmed by ileocolonoscopy (endoscopic Mayo score ≥1) with colonic biopsies of inflamed mucosa [24].

Endoscopic images and written reports from initial and follow-up ileocolonoscopy were re-assessed for severity of disease using the Mayo score (0–3 scale). We favored the use of a histological score, rather than the endoscopy score, in order to classify the severity of disease because of the high subjectivity of retrospective re-assessment of endoscopic images using the Mayo score. Next to this subjectivity, older endoscopic images are of poorer quality, and thus insufficient for reliable re-assessment. Histopathological assessment of biopsies was performed independently by two reviewers, both blinded for patient characteristics, by using the Geboes score to classify the histologic type and severity of disease, where Grade 0 indicates architectural changes, Grade 1 chronic inflammatory infiltrate, Grade 2 eosinophils (2A) and neutrophils (2B) in the lamina propria, Grade 3 neutrophils in the epithelium, Grade 4 crypt destruction and Grade 5 indicates the presence of erosions and ulcerations in biopsy specimens. Within each Grade (0–5), the stage of severity is then classified in stages arranging from 0 to 3 (no abnormalities to severe changes) [25]. UC patients with a Geboes score ≥ 3.1 (meaning a score higher than stage 1 of Grades 3 or 4 or 5) were identified as patients with histological active disease. Patients were excluded if paraffin-embedded biopsies at baseline and/or follow-up ileocolonoscopy were unavailable for investigation. Follow-up ileocolonoscopy was performed based on routine clinical care, i.e., in case of symptoms suggestive of exacerbation or when surveillance was indicated. The follow-up period was defined as the time between initial diagnosis and the last visit at the outpatient clinic, date of death or loss to follow-up due to migration.

The following variables were extracted from the medical records: patient demographics, disease phenotype (including disease location and behavior) at initial diagnosis according to the Montreal classification [26], the first effective remission-induction therapy following initial presentation, time between initial presentation and first exacerbation (defined by a combination of clinical symptoms, step-up in therapy, biochemical activity and/or confirmed endoscopic disease activity), response to anti-TNF treatment, the need for a subtotal colectomy and medication use at the time of follow-up ileocolonoscopy (i.e., aminosalicylates, topical steroids, systemic steroids, immunomodulators or biologicals).

Healthy controls (HC) who underwent ileocolonoscopy for polyp surveillance or iron deficiency from whom paraffin embedded mucosal biopsies were available were included in the control group. Histopathological analysis confirmed the absence of architectural changes, chronic inflammatory infiltrate, influx of neutrophils and eosinophils in the lamina propria, neutrophils in the epithelium, crypt destruction and erosions or ulcerations in HC.

### 2.2. Immunohistochemistry 

3μm-thick sections from formalin-fixed, paraffin-embedded archived blocks of biopsied specimen of colonic mucosa were cut and selected for immunohistochemistry. Besides hematoxylin and eosin (HE, from Klinipath) staining, immunostaining was performed with the following monoclonal antibodies: CD3 (from Novocastra, clone LN10, 1:50, marking T cells), CD20 (from Thermo scientific, clone L-26, 1:125, marking B cells), ERG (from Ventana, clone EPR3864, demonstrating only nuclei of endothelial cells on all blood vessels), MECA-79 (from Santa cruz Biotechnology, clone MECA-79, 1:350, demonstrating 6-sulpho-sialyl Lewis on core-1 branched O-linked sugars (PNAd)), MECA-376 (from Hycult Biotech, clone 314G8, 1:50, marking MAdCAM-1). Slides were incubated with these antibodies in an automatic immunostainer (Ventana Benchmark Ultra, Eindhoven, the Netherlands). After performing immunostaining, these slides were scanned with an Intellisite high-resolution scanner (Philips ultra-fast scanner 1.6 RA; Philips Digital Pathology, Best, The Netherlands) and analyzed within the IntelliSite Pathology Solution Image Management System (IMS, Philips Digital Pathology, Best, The Netherlands).

The most inflamed biopsy at each HE stained slide was selected using the Geboes score. In each patient, the surface of the most affected colonic biopsy slide and the area of follicular tissue was determined on the same HE, CD3 and CD20 immunostained sections. The total surface area of the biopsy (including follicular and extrafollicular tissue) was measured in square millimetres (mm^2^). Lymphoid follicles were optically counted and divided in primary and secondary follicles based on the presence or absence, respectively, of germinal centers besides B and T cell compartments. Absolute numbers of ERG^+^, PNAd^+^ and MAdCAM-1^+^ venules were optically counted within the extra- and intra-follicular surface of the whole biopsy at 40x magnification by one analyst. The proportion of PNAd^+^ and MAdCAM-1^+^venules among all ERG^+^venules was displayed in percentages (% PNAd^+^ venules/ERG^+^venules and % MAdCAM-1^+^ venules/ERG^+^venules). We also described the density of PNAd^+^ and MAdCAM-1^+^ venules computed by dividing the absolute number of venules by the biopsy surface (respectively PNAd^+^ venules/mm^2^ and MAdCAM-1^+^ venules/mm^2^) and the percentage of PNAd^+^venules also expressing MAdCAM-1^+^ (% double positive venules/ERG^+^venules).

### 2.3. Multiplex Immunoassay

Additionally, we aimed to analyze CCL-19 and CXCL-13 in the local and systemic cytokine milieu of UC patients. CCL-19 is expressed on HEVs and in T-cell zones of SLOs and it is required for trafficking and positioning of T cells and dendritic cells within SLOs. CXCL-13, or BLC/BCA-1, is a chemokine expressed in B-cell follicles. It is suggested that these homeostatic chemokines (both CCL-19 and CXCL-13) contribute to the generation and organization of lymphoid neogenesis [12].

Following initial ileocolonoscopy, venous blood was obtained from 22 UC patients and 10 HC. Colonic biopsies taken during endoscopy from newly diagnosed UC patients and HC were kept in a phosphate-buffered saline solution at 2–8 °C and processed within eight hours. Specimens were pooled and finely minced in Hanks’/1% bovine serum albumin using a 70-mm gaze and spatula followed by Ficoll density gradient centrifugation. The single cell suspension was resuspended, after washing, and stimulated overnight with PMA/ionomycin to induce chemokine/cytokine excretion.

Multiplex immunoassays were performed at the MultiPlex Core Facility of the Center for Translational Immunology (UMC Utrecht, Utrecht, The Netherlands) using an in-house validated platform (ISO9001) on the serum samples and supernatants of the cultured biopsies as previously described [27]. Undetectable analyte results were replaced by the lowest measured value in the patient group divided by two. This surrogate value was always below the lower limit of detection.

### 2.4. Statistical Analysis 

Continuous variables were expressed as mean with standard deviation (±SD) or as median with interquartile range (IQR) depending on skewness. Categorical data were expressed as numbers with percentage and analyzed using the Chi-square when necessary. To study change in expression of PNAd and MAdCAM-1 on colonic mucosal venules over time, paired baseline and follow-up variables were compared using the paired T-test or Wilcoxon signed rank test depending on skewness, whereas continuous variables of unpaired groups were compared using the independent T-test or Mann-Whitney U test.

To determine if the variables age, gender, smoking behavior, symptom duration prior to initial diagnosis, location of UC, disease activity expressed using the Geboes score (severity stage for each separate Grade), effective remission induction treatment directly after diagnosis and treatment at follow-up were correlated with the expression of PNAd and MAdCAM-1 on colonic mucosal venules at diagnosis and during follow-up, a linear regression model was used. Factors with a *p*-value <0.2 in univariable analysis were included in a multivariable linear regression model with backward elimination. A two-sided p-value of 0.05 was considered to be statistically significant. Data analysis was performed using the SPSS statistical software (version 24.0.0.0; IBM Corp, Armonk, NY, USA) and GraphPad Prism (Graphpad Software version 7.0, La Jolla, CA, USA).

### 2.5. Ethics

The study protocol (NL28761.091.09) was approved by the research ethics committee of the Radboud University Nijmegen Medical Centre (CMO Regio Arnhem-Nijmegen, Nijmegen, The Netherlands). The procedures were performed in accordance with the Declaration of Helsinki (version 9, 19 October 2013).

## 3. Results

### 3.1. Baseline Characteristics Study Population

We included biopsy specimens of 378 untreated UC patients at diagnosis and during follow-up, and 10 HC. All patients had an endoscopic Mayo score of at least one at baseline colonoscopy. To reach remission after initial diagnosis, 223 (59%) UC patients required aminosalicylates, 42 (11.1%) needed topical steroids and the remaining patients were in need of oral steroids, anti-TNF treatment or resective surgery (*n* = 104, 27.6%). The baseline characteristics of all patients and HC are presented in Table 1.

### 3.2. The Presence of PNAd^+^ and MAdCAM-1^+^ Venules in Colonic Biopsies at Baseline 

In the colonic mucosa of UC patients, higher numbers of follicles/mm^2^ (median: 1.3 (IQR 0.7–2.1)) were present compared to HC (median: 0.04 (IQR 0.0–0.09), *p* = 0.001) (Table 2). Consequently, the total follicular surface per colonic biopsy was higher in UC (median 0.15 (0.05–0.32) mm^2^) vs. HC (0.04 (0.0–0.09) mm^2^, *p* = 0.004). All ERG^+^ venules were located outside these lymphoid follicles (i.e., extrafollicular). Likewise, the majority of PNAd^+^ and MAdCAM-1^+^ venules were found extrafollicular (Figure 1).

At diagnosis, the median proportion of PNAd^+^ venules was 4.9% (IQR 2.0–8.3%) in biopsies of UC patients with only 15 patients having none PNAd^+^ venules. These PNAd^+^ venules were completely absent in all HC (0.0 % (0.0–0.0%), *p* = 0.001) (Figure 1). The median proportion of MAdCAM-1^+^ venules was also higher in UC patients at diagnosis (5.8% (IQR 2.5–9.9%)) compared to HC (0.8% (IQR 0.0–3.8%), *p* = 0.001). Of all ERG^+^ venules, 1.6% (IQR 0.4–3.7%) were both MAdCAM-1^+^ and PNAd^+^, whereas 48.0% (IQR 13.9–69.4%) of all PNAd^+^venules were MAdCAM-1^+^ as well. A correlation was found between the number of follicles per mm^2^ biopsy and the relative number of PNAd^+^venules (Figure 2, *Rho* = 0.512, *p* = 0.001).

The median number of follicles/mm^2^ measured at follow-up endoscopy in UC patients in endoscopic remission was significantly lower (0.4 (IQR 0.0–0.7)) compared to their own baseline values at diagnosis (1.1 (0.5–1.9), p = 0.001) resulting in a smaller total follicular surface (0.03 (IQR 0.0–0.1) mm^2^ versus the median: 0.12 (0.03-0.3) mm^2^, p = 0.001). The proportion of PNAd^+^ venules in UC patients in endoscopic remission completely disappeared (0.0%(IQR 0.0–0.08%), Figure 3). In contrast, the proportion of MAdCAM-1^+^venules remained the same when comparing patients in endoscopic remission at follow-up (7.8% (IQR 4.5–11.8%)) with their own baseline levels (6.9% (IQR 2.6–11.0%), p = 0.15). 

UC patients with active endoscopic disease at follow-up demonstrated significant lower proportions of PNAd^+^venules (4.0% (IQR 1.3–7.8%)) compared to their own baseline levels (5.3% (2.2–8.3%), p = 0.04), whereas an upregulation of MAdCAM-1 from baseline (5.5% (IQR 2.5–9.5%)) to follow-up (7.8% (IQR 4.5–12.0%), p = 0.001) was demonstrated. No difference in the proportion of MAdCAM^+^ venules was detected between patients in endoscopic remission and active disease during follow-up (p = 0.80) (Table 3).

### 3.3. Association of PNAd and MAdCAM-1 with Disease Phenotype at Diagnosis

The results of the linear regression analysis are shown in Table 4. The univariable analysis showed that male sex (*β* = 0.16, SE 0.006, *p* = 0.002), smoking (*β* = 0.13, SE 0.004, *p* = 0.01) and pancolitis (*β* = 0.35, SE 0.004, *p* = 0.001) were significantly associated with lower baseline proportions of PNAd^+^ venules in UC patients (Table 4A). Of note, the absolute number of PNAd^+^ venules was lower in smoking UC patients, while the number of ERG^+^ venules was comparable between smokers and non-smokers (*p* = 0.001). 

In addition, a longer duration of symptoms before diagnosis (*β* = 0.25, SE 0.001, *p* = 0.001), histological disease activity, as indicated by the Geboes scores describing architectural changes (Geboes 0, *β* = 0.20, SE 0.004, *p* = 0.002), chronic inflammatory infiltrate (Geboes 1, *β* = 0.22, SE 0.006, *p* = 0.001) and the presence of erosions and ulcerations (Geboes 5, *β* = 0.13, SE 0.002, *p* = 0.05), and rectal disease activity (*β* = 0.35, SE 0.004, *p* = 0.001) were significantly associated with higher baseline proportions of PNAd^+^ venules in UC (Appendix A). The age at diagnosis and Geboes scores describing the amount of eosinophils in the lamina propria (2A) and the presence of cryptitis (4) were not significant variables in univariable analysis, but were included in the multivariable analysis as the p-value was below the prespecified threshold of 0.2. There was no association between Geboes describing the number of neutrophils in the lamina propria (2B) and epithelium (3) and the proportion of PNAd^+^ venules. Multivariable analysis identified symptom duration before diagnosis (*p* = 0.01), disease location (*p* = 0.001) and Geboes 0 (*p* = 0.01) as independent variables associated with the proportion PNAd^+^ venules. A follow-up endoscopy, disease location was also an independent factor associated with the proportion of PNAd^+^ venules (*p* = 0.001, Appendix A).

All subdomains of the Geboes score were associated with lower baseline proportions of MAdCAM-1^+^ venules in UC patients with p-values below 0.2 in the univariable analysis (Table 4B). In multivariable analysis, this association was not statistically significant.

### 3.4. Association of PNAd^+^ and MAdCAM-1^+^ Venules with Disease Course

Patients who responded to aminosalicylates and/or topical steroids (*n* = 265, 70%) as initial remission induction therapy displayed a higher proportion of PNAd^+^ venules in baseline biopsies compared to patients in need of oral steroids, anti TNF treatment or surgery (*n* = 104, 27.6%, *p* = 0.02). The proportion of MAdCAM-1^+^ venules at baseline did not distinguish responders to aminosalicylates and/or topical steroids from responders to oral steroids, anti-TNF and/or surgery.

Furthermore, in patients with active disease at time of follow-up endoscopy, we studied the association between current treatment and the proportion of PNAd^+^ and MAdCAM-1^+^ venules. The use of oral steroids and/or anti-TNF during follow-up was associated with a significantly lower proportion of PNAd^+^ venules compared to patients with active disease who did not use any medication at that time. In addition, a significant decline in the proportion of MAdCAM-1^+^ venules was found in patients using oral steroids during follow-up. Treatment with Vedolizumab was associated with an increased proportion of MAdCAM-1^+^ venules in active disease during follow-up (Figure 4).

After initial remission, 89% of UC patients (*n* = 337) experienced at least one relapse, while 11% (*n* = 41) did not experience an exacerbation. The median number of clinical, biochemical and/or endoscopic confirmed exacerbations was two (IQR 1–3) during 128 months (IQR 85–165) of follow-up (which is equal to 0.18 (IQR 0.09–0.3) exacerbations per patient-year). The first exacerbation occurred after a median period of 25 months (IQR 10–59). When comparing patients with the highest proportion MAdCAM-1^+^ venules (75th percentile, ≥ 9.9%) with patients with the lowest proportions (25th percentile, ≤ 2.5%) at baseline endoscopy, we found that patients with initial higher numbers of MAdCAM-1^+^ venules experienced significantly more exacerbations per year (0.24 (IQR 0.15–0.33) versus 0.17(0.08–0.32), *p* = 0.03). During a first exacerbation, the proportion of MAdCAM-1^+^ venules was higher compared to baseline levels (respectively 7.5% (4.4–10.9%) and 5.8% (2.6–10.0%), *p* = 0.001). In case of a following exacerbation, the proportion of MAdCAM-1^+^ venules (8.5% (5.0–12.5%)) was even higher compared to the proportions during the first exacerbation (Figure 5). In contrast, there was no upregulation of proportions PNAd^+^ venules during subsequent flares. 

### 3.5. CXCL-13 and CCL-19 in Serum and Stimulated Biopsies

In 22 of the included UC patients and 10 HC, the chemokines CXCL-13 and CCL-19 were analyzed in the serum and supernatants of stimulated biopsies which were collected during baseline endoscopy. In the serum of UC patients, we demonstrated statistically significant higher levels of CXCL-13 and CCL-19 compared to HC (CXCL-13: 86.2 pg/mL (IQR 58.3–109.4) versus 36.9 (IQR 31.3–48.8), *p* = 0.001; CCL-19: 167.7 pg/mL (IQR 133.8–204.3) versus 100.3 (IQR 37.4–158.1), *p* = 0.02). In the supernatant of the stimulated biopsies from UC patients, statistically significant higher levels of CCL-19 were found compared to HC (16.4 pg/mL (IQR 13.4–17.2) versus 10.5 (IQR 9.4–12.9), *p* = 0.001). The levels of CXCL-13 in the supernatant were also higher in UC patients compared to HC, but without reaching statistical significance (Table 5).

The number of follicles/mm^2^ biopsy was correlated with the CXCL-13 level in serum (*Rho* = 0.47, *p* = 0.01) and the CCL-19 level in the supernatant (*Rho* = 0.43, *p* = 0.02), suggesting a potential role for these chemokines in TLO formation. The density of MAdCAM-1^+^ venules was related to the amount of CCL-19 in the supernatant (*Rho* = 0.53, *p* = 0.003). CCL-19 in the supernatant also correlate with the absolute number and density of PNAd^+^ venules (per mm^2^) (*Rho* = 0.41, *p* = 0.03 and *Rho* = 0.39, *p* = 0.04). 

## 4. Discussion 

In a large cohort of untreated UC patients at diagnosis we found a high proportion of colonic mucosal PNAd^+^ and MAdCAM-1^+^ venules (nearly all located outside lymphoid follicles). In HC, PNAd^+^ venules were completely absent and MAdCAM-1^+^ venules were present in very low numbers. The proportion of PNAd^+^ HEVs was correlated with more follicles and a greater follicular surface. PNAd^+^ HEVs were absent in the colonic mucosa of UC patients when no inflammation was present at follow-up. Simultaneously, in remission the number of follicles shifted to lower levels comparable to HC, while the proportion of MAdCAM-1^+^ venules remained unchanged. On the other hand, patients with active endoscopic disease at follow-up were found to have a comparable proportion of PNAd^+^ HEVs to levels at diagnosis, while a further increase in the proportion of MAdCAM-1^+^ venules was found with each subsequent exacerbation.

In line with our results, several reports have shown that PNAd^+^ HEVs are never found in the lamina propria of the intestine of healthy controls [27,28]. These PNAd^+^ HEVs were only found in Peyers patches, the appendicular mucosa and in mesenteric lymph nodes, but with a less intense staining compared to PNAd^+^ HEVs found in tonsils and peripheral lymph nodes [27,28]. 

Our findings of PNAd^+^ venules being present in the colonic mucosa during inflammation of UC are in line with two earlier studies performed in a small patient group [14,15]. However, these studies did not take MAdCAM-1 positivity into account, and were cross-sectional in design providing no information during the follow-up of the disease. Expression of PNAd on mucosal HEVs in active disease enables the recruitment of T cells to TLOs [11]. Results from the previous studies using different methods (flowcytometry versus immunohistochemistry) suggest the recruitment of L-selectin-expressing naive T cells through these PNAd^+^ HEVs [14,15]. Categorizing patients into groups based on the relative number of PNAd^+^ venules (HEV^high^ and HEV^low^), a correlation was found between the presence of more lymphoid follicles and higher percentages of Tn and Tcm lymphocytes in the HEV^high^ group [14]. In the present study, we demonstrated a statistically significant correlation between the proportion of PNAd^+^ venules and the number of lymphoid follicles. In HC, the PNAd^+^ venules are exclusively found in SLOs and absent in extrafollicular mucosal tissue. The presence of these extrafollicular PNAd^+^ HEVs in active UC suggests formation of newly formed TLO in the chronic inflamed tissue thereby controlling the influx and potential local activation of T cells. We demonstrate that PNAd^+^ HEVs disappear in UC patients with inactive disease at follow-up endoscopy. Therefore, these venules can be described as a marker of disease activity in UC.

In HC, MAdCAM-1 is exclusively expressed on endothelium of flattened venules in the lamina propria along the gastrointestinal tract and in associated lymphoid tissue, and it is not detected in extra-intestinal tissues, including those with mucosal surfaces [8]. This is in line with our results in the biopsies of HC. The available literature in long-standing UC patients describes increased numbers of MAdCAM-1^+^ venules compared to the healthy control [8,16,29,30,31]. We demonstrated comparable increased numbers of MAdCAM-1^+^ venules in newly diagnosed UC. In accordance with the present results, other studies showed no statistically significant difference between arbitrary chosen active and remission phases for the number of MAdCAM-1^+^ venules in different UC patients during follow-up. We describe a statistically significant upregulation of the proportion of MAdCAM-1^+^ venules over time, from baseline to the first exacerbation within the same patient. After experiencing several exacerbations, the number of MAdCAM-1^+^ venules increase to even higher levels compared to those found at the moment of the first exacerbation. The MAdCAM-1 levels in patients who never experienced an exacerbation after diagnosis remain comparable to levels at diagnosis. In primary cultures of human intestinal microvascular endothelial cells, MAdCAM-1 gene and protein expression is induced in response to inflammatory cytokines (TNFα and IL-1β) and bacterial endotoxin (LPS) requiring both NF-κB and PI3-K/Akt activation [32]. Longer culture duration and higher cellular densities of microvascular endothelial cells (suggesting that cell-cell interaction plays a critical role) lead to more MAdCAM-1 expression induced by TNFα or LPS stimulation [32]. A longer total disease duration (time of all exacerbations added together) resulted in more exposure to these inflammatory cytokines and can possibly explain the upregulation of MAdCAM-1 over time after experiencing more exacerbations.

Besides HEVs and flattened venules, different chemokines and cytokines play an important role in controlling the migration of T cells to the intestine. Homeostatic chemokines such as CXCL-13 and CCL-19 and lymphoneogenic cytokines like lymphotoxin αβ are involved in the recruitment of peripheral T cells to the site of inflammation where newly developed tertiary lymphoid organs are formed [12,33]. In the serum and supernatant of stimulated biopsies from 22 UC patients we found an upregulation of both CXCL-13 and CCL-19 compared to healthy controls. Higher levels of CXCL-13 in the serum and CCL-19 in the supernatant of biopsies were correlated with the presence of more follicles/mm^2^ in the inflamed gut mucosa. Higher densities of PNAd^+^ HEVs and MAdCAM-1^+^ venules were also associated with higher CCL-19 in the supernatant. These findings suggest a possible role of CXCL-13 and CCL-19 in PNAd^+^ and MAdCAM-1^+^ expression and TLO formation [11,12,33,34,35]. The use of these chemokines as targets may be limited because similar molecular mechanisms are involved in the formation and maintenance of secondary and tertiary lymphoid organizations. Blocking CXCL-13 or CCL-19 could potentially harm lymphoid follicles disseminated throughout the whole body [35].

The presence of large aggregates of lymphocytes with T/B cell compartmentalization and PNAd^+^ HEVs (indicating lymphoid neogenesis) in the synovial fluid of patients with rheumatoid arthritis (RA) has been associated with disease severity, the frequency of exacerbations and an inferior response to frontline biological therapies that target TNF [36]. Therapeutic reversal of these lymphoid neogenesis features was correlated with good clinical responses in RA, but it remains unclear whether newly formed lymphoid follicles (TLOs) indicate distinct disease phenotypes or an evolutionary manifestation of chronic inflammation [37]. In our UC cohort, we found that higher proportions of PNAd^+^ venules at baseline endoscopy were associated with longer symptom duration before diagnosis, histologic more severe disease activity as indicated by the Geboes score describing architectural changes and rectal disease activity. The correlation of higher proportions of PNAd^+^ HEVs with longer duration of complaints before diagnosis suggests that PNAd^+^ HEVs are a reflection of chronicity. Lower proportions of PNAd^+^ venules were correlated with male sex, smoking and extensive colitis. The correlation between higher numbers of PNAd^+^ HEVs in baseline biopsies and initial response on 5-ASA therapy and local steroids in our retrospective cohort should be interpreted with caution. At this point, a relation between PNAd expression and the working mechanism of 5-ASA therapies on mucosal inflammation is not known. The effect of medication and its relation to PNAd expression should be further studied using a prospective study design. When comparing our results in UC patients with findings in RA, we can confirm a similar role for PNAd^+^ HEVs in assessing disease activity.

A higher proportion of MAdCAM-1^+^ venules at baseline was associated with more exacerbations during clinical follow-up. Given the upregulation of MAdCAM-1 over time, based on the number of exacerbations, this supports the hypothesis that treatment with antibodies against MAdCAM-1(Ontamalimab) could be effective in inducing and maintaining remission in patients with longstanding disease (activity) [19].

It is well known that smoking ameliorates UC and smoking cessation has been associated with the onset of UC [38,39]. Nicotine has been considered to be responsible for the protective effect of smoking in UC, while the exact mechanism of action remains unclear. The research group of Maruta demonstrated in vivo that nicotine suppresses the increased recruitment of leukocytes. This was explained by a decrease of MAdCAM-1 expression on endothelial cells in the gut of DSS-induced colitis, possibly through a direct effect of nicotine on the vascular endothelium [40,41]. In contrast, our results indicate that smoking leads to reduced proportions of PNAd^+^ venules with no difference in the numbers of MAdCAM-1^+^ venules between smokers and non-smokers. The reduction in PNAd^+^ venules can lead to a decreased influx of naive T cells to the gut, which has been associated with less severe inflammation [42].

At the moment, there is an urgent need for biomarkers predicting response prior to treatment initiation as different treatment options are not effective in all patients [43]. For instance, pre-treatment analysis of α4β7 expression on lymphocytes in blood of IBD patients could be of value in predicting response on anti-α4β7 antibodies [44,45]. We showed that PNAd^+^ venules contribute in predicting response on initial induction therapy. Since PNAd^+^ venules are also presented in lymphoid tissue outside the gastrointestinal tract, blocking them might have harmful side effects [46]. UC patients with higher numbers of PNAd^+^ HEVs, associated with more T_n_- and fewer α4β7^+^ T_em_ cells [14], might benefit less of treatment with anti-α4β7^+^ antibodies and/or anti-MAdCAM-1 treatment. MAdCAM-1^+^ venules might be a better therapeutical target compared to PNAd^+^ venules due to their gut-specific presence and upregulation in time during active inflammation. Patients with higher numbers of MAdCAM-1^+^ venules, suggesting more recruitment of α4β7^+^ T cells, might benefit more from anti-MAdCAM-1 treatment.

The strength of the present study is that it comprises a large cohort of newly diagnosed UC patients with serial measurements during remission and active disease phases, revealing the evolution in time of these vascular adressins. Further prospective research should be undertaken to investigate the value of PNAd^+^ and MAdCAM-1^+^ venules as treatment targets, biomarkers in predicting response to therapy and the effect of different treatments on the development and function of these venules. 

In conclusion, a new formation of mucosal PNAd^+^ HEVs is associated with the presence and histologic severity of inflammation and therefore a potential marker for disease activity in UC. Higher proportions of PNAd^+^ HEVs were associated with more colonic follicles, suggesting formation of tertiary lymphoid organs in the inflamed gut mucosa. As a high proportion of MAdCAM-1^+^ venules is predictive for the number of exacerbations and increases with each subsequent exacerbation, prospective studies should focus on its possible predictive role and value as a therapeutic target.

## Figures and Tables

**Figure 1 cells-09-00891-f001:**
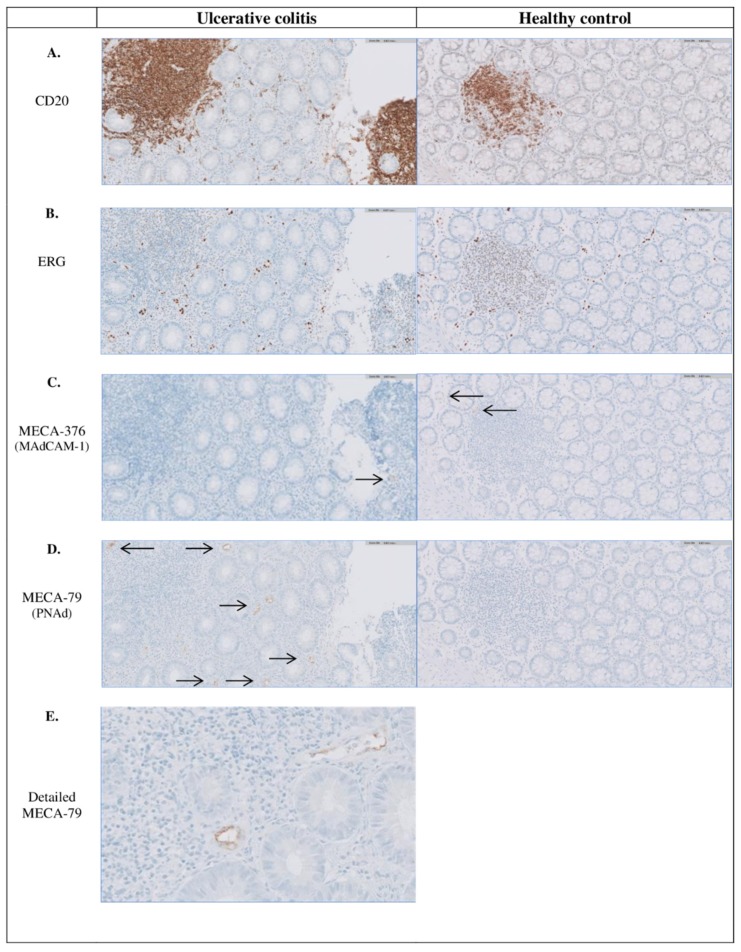
Immunohistochemical staining indicating the presence of follicles and PNAd^+^ and MAdCAM-1^+^ venules in inflamed colonic tissue of UC patients and non-inflamed colonic tissue of a healthy control. Representative photomicrographs, with a magnification of x 20, of a colonic biopsy from a UC patient with (**A**) CD20 staining indicating B-cells (and follicles), (**B**) ERG staining indicating all venules, (**C**) MECA-367 staining indicating MAdCAM-1^+^ venules (pointed out with black arrows), (**D**) MECA-76 staining indicating PNAd^+^ venules (pointed out with black arrows) and (**E**) MECA-76 staining in detail with a magnification of x 40. Almost all PNAd^+^ and MAdCAM-1^+^ venules are located extrafollicular. MAdCAM-1, mucosal vascular addressin cell adhesion molecule-1; PNAd, peripheral node addressin; UC, Ulcerative colitis.

**Figure 2 cells-09-00891-f002:**
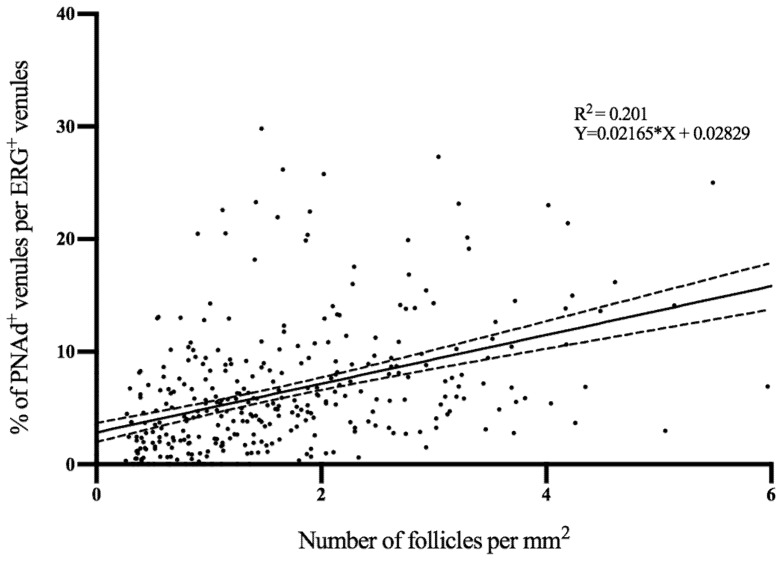
Correlation between the number of follicles per mm^2^ and the proportion of PNAd^+^ venules per ERG^+^ venules in the biopsies of inflamed UC. ERG, ETS related gene; PNAd, peripheral node addressin; UC, ulcerative colitis.

**Figure 3 cells-09-00891-f003:**
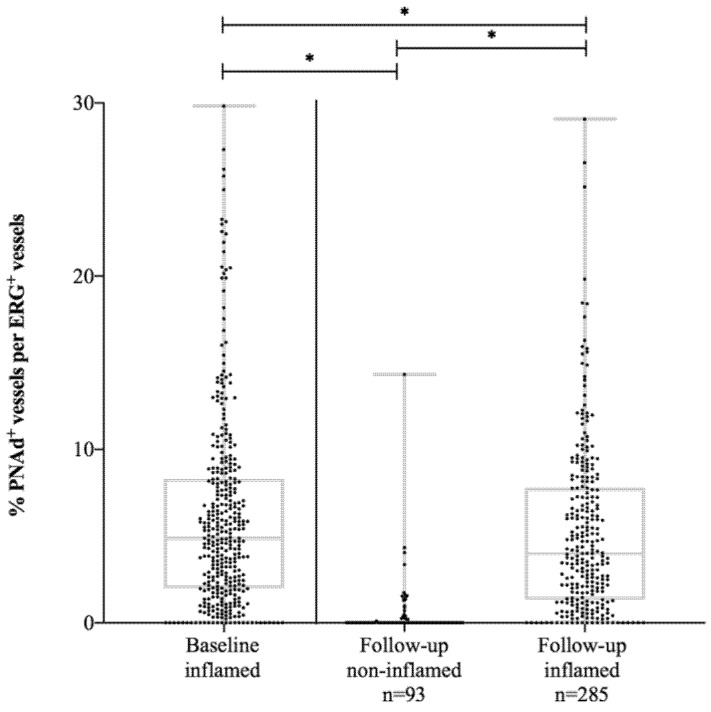
The proportion of PNAd^+^ vessels per ERG^+^ vessels in UC patients at baseline and at follow-up. PNAd, peripheral node addressin; UC, ulcerative colitis. * p-value < 0.05.

**Figure 4 cells-09-00891-f004:**
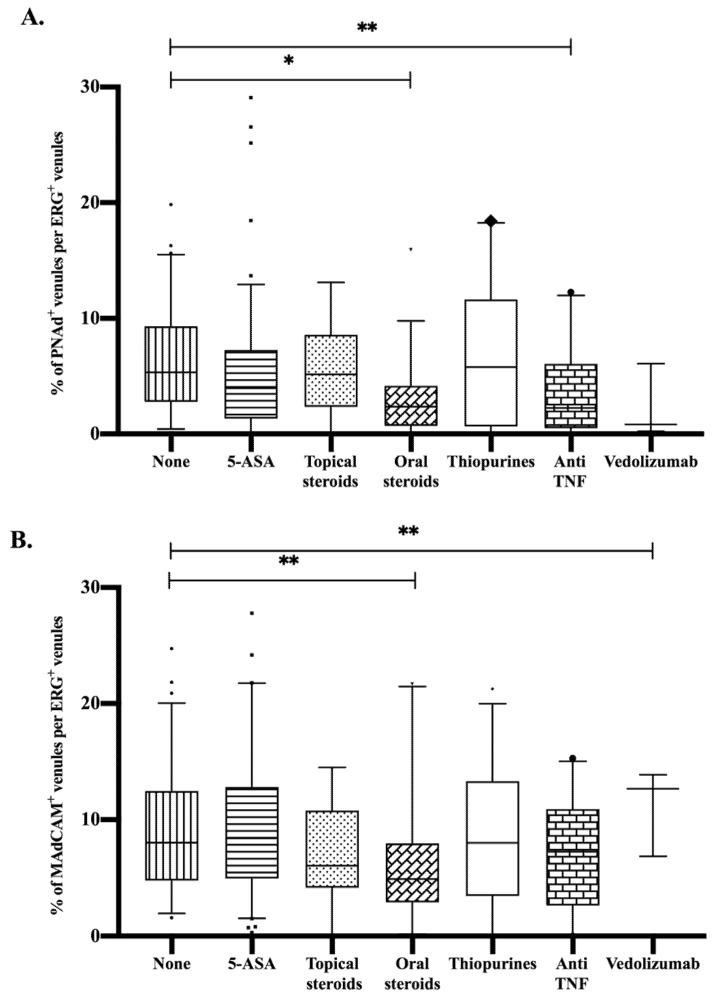
The effect of treatment during follow-up endoscopy on the percentages of (**A**) PNAd^+^ venules and (**B**) MAdCAM-1^+^ venules per ERG^+^ venules.* *p*-value < 0.001; ** *p*-value < 0.05; 5-ASA, aminosalicylates; MAdCAM-1, mucosal vascular addressin cell adhesion molecule-1; PNAd, peripheral node addressin; UC, ulcerative colitis.

**Figure 5 cells-09-00891-f005:**
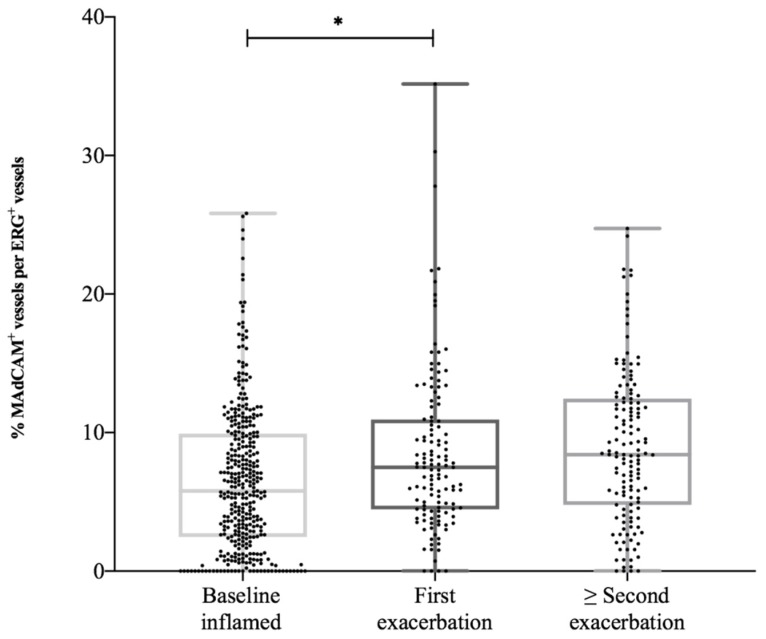
The upregulation of the proportion of MAdCAM-1^+^ vessels per ERG^+^ vessels in UC patients during exacerbations. MAdCAM-1, mucosal vascular addressin cell adhesion molecule-1; UC, ulcerative colitis. * *p*-value < 0.05.

**Table 1 cells-09-00891-t001:** Baseline characteristics.

	UC (*n* = 378)	HC (*n* = 10)
**Age at Diagnosis in Years **	43 (30–57)	36 (26–43)
Sex		
- Female	190 (50.3%)	6 (60%)
- Male	188 (49.7%)	4 (40%)
**Extra-Intestinal Manifestations**		
- Yes	46 (12.2%)	-
**Family History of IBD**		
- Yes	42 (11.1%)	-
**Duration of Complaints before Diagnosis in Weeks**	10 (4–16)	-
**Smoking Status at Baseline Endoscopy**		
- Ceased	84 (21.5%)	0 (0%)
- Yes	48 (12.3%)	3 (30%)
- No or unknown	246 (66.2%)	7 (70%)
**Calprotectin**	322 (167–1161)	-
**Mayo Endoscopic Score**		
- Mayo 0	-	10 (100%)
- Mayo 1	156 (41.3%)	0 (0%)
- Mayo 2	174 (46.0%)	0 (0%)
- Mayo 3	48 (12.7%)	0 (0%)
**UC Localization**		
- Extent		
○ E1: Ulcerative proctitis	136 (36.0%)	-
○ E2: Left-sided UC	149 (39.4%)	
○ E3: Extensive UC	93 (24.6%)	
**Histological Inflammation Geboes Score **		
- ≥ 0.1 and < 3.1	2 (0.5%)	10 (100%)
- ≥ 3.1	376 (99.5%)	0 (0%)
**Location Biopsies Taken at Baseline Endoscopy**		
- Rectum	143 (37.8%)	-
- Sigmoid	152 (40.2%)	-
- Right-sided	2 (0.5%)	-
- Left- and right-sided	81 (21.4%)	10 (100%)
**Effective Remission Induction Treatment at Diagnosis**		
- No treatment	9 (2.4%)	10 (100%)
- 5-ASA	223 (59.0%)	-
- Topical steroids	42 (11.1%)	-
- Oral steroids	86 (22.8%)	-
- Anti-TNF	15 (4.0%)	-
- Resective surgery	3 (0.8%)	-

Values expressed in n (%) or as median with interquartile range; HC, Healthy control; IBD, Inflammatory Bowel Diseases; IQR, Interquartile range; UC, ulcerative colitis.

**Table 2 cells-09-00891-t002:** The presence of follicles and extrafollicular PNAd^+^, MAdCAM-1^+^ and/or ERG^+^ venules in the inflamed colonic biopsies at UC diagnosis compared to healthy controls.

	UC (Inflamed) *n* = 378	HC (Non-Inflamed) *N* = 10	*P*-Value
**Total Surface of the Biopsy in mm^2^**	2.3 (1.8–3.0)	2.1 (1.6–3.1)	*0.62*
**Total Follicular Surface in mm^2^**	0.15 (0.05–0.32)	0.04 (0.0–0.09)	0.004
**Total Extra Follicular Surface in mm^2^**	2.0 (1.5–2.7)	2.0 (1.6–3.1)	0.93
**Follicles**			
- Absolute number	3.0 (1.0–5.0)	1.0 (0.0–1.0)	0.001
- Number per mm^2^	1.3 (0.7–2.1)	0.04 (0.0–0.09)	0.001
**Extrafollicular ERG^+^ Venules**			
- Absolute number	366 (263–485)	206 (162–244)	0.001
- Number per mm^2^ (density)	159 (130–193)	108 (66–116)	0.001
**Extrafollicular PNAd^+^Venules**			
- Absolute number	17.5 (6.0–34.0)	0.0 (0.0–0.0)	0.001
- Number per mm^2^ (density)	7.7 (2.6–14.0)	0.0 (0.0–0.0)	0.001
- As % of ERG^+^ venules (proportion)	4.9 (2.0–8.3)	0.0 (0.0–0.0)	0.001
**Absolute Number of Intrafollicular PNAd^+^ Venules**	1.0 (0.0–4.0)	0.0 (0.0–0.0)	0.008
**Extrafollicular MAdCAM-1^+^Venules**			
- Absolute number	19.0 (8.0–36.0)	1.5 (0.0–8.8)	0.001
- Number per mm^2^ (density)	9.1 (3.8–15.6)	0.5 (0.0–3.9)	0.001
- As % of ERG^+^ venules (proportion)	5.8 (2.5–9.9)	0.8 (0.0–3.8)	0.001
**Absolute Number of Intrafollicular MAdCAM-1^+^ Venules**	0.0 (0.0–3.0)	0.0 (0.0–1.5)	0.19
**Extrafollicular MAdCAM-1^+^ PNAd^+^Venules**			
- Absolute number	6 (1.0–15.0)	0 (0–0)	0.001
- Number per mm^2^ (density)	2.4 (0.6–6.0)	0 (0–0)	0.001
- As % of ERG^+^ venules (proportion)	1.6 (0.4–3.7)	0 (0–0)	0.001
**Percentage MAdCAM-1^+^ of PNAd^+^ Venules**	42.9 (18.1–70)	*	-

**Table 3 cells-09-00891-t003:** The presence of follicles and (PNAd^+^, MAdCAM-1^+^, ERG^+^) venules in inflamed and non-inflamed colonic biopsies of UC patients at follow-up endoscopy.

	UC Patients in Remission at Follow-Up *n* = 93			UC Patients with Active Disease at Follow-Up *n* = 285			
	Baseline	Follow-up	*P*-Value	Baseline	Follow-up	*P*-value	*P*-value
	Inflamed	Non-Inflamed		Inflamed	Inflamed		Follow-Up Non-Inflamed vs Inflamed
**Total Surface of the Biopsy in mm^2^**	2.4 (1.7–3.2)	2.0 (1.5–2.7)	*0.1*	2.3 (1.8–2.9)	2.2 (1.7–2.8)	0.21	0.39
**Total Follicular Surface in mm^2^**	0.12 (0.03–0.3)	0.03 (0.0–0.1)	*0.001*	0.16 (0.06–0.34)	0.1 (0.04–0.3)	0.001	0.001
**Total Extra Follicular Surface in mm^2^**	2.1 (1.5–3.0)	2.0 (1.5–2.6)	*0.43*	2.0 (1.6–2.6)	2.0 (1.6–2.6)	0.92	0.6
**Follicles**							
- Absolute number	2 (1–4)	1 (0–1)	*0.001*	3 (1–5)	2 (1–4)	0.001	0.001
- Number per mm^2^	1.1 (0.5–1.9)	0.4 (0.0–0.7)	*0.001*	1.4 (0.7–2.2)	1.1 (0.5–1.8)	0.001	0.001
**ERG^+^ Venules**							
- Absolute number	336 (232–468)	235 (164–339)	*0.001*	371 (274–490)	394 (292–546)	0.12	0.001
- Number per mm^2^ (density)	146 (123–173)	123 (93–155)	*0.001*	163 (131–194)	184 (150–224)	0.001	0.001
**Extrafollicular PNAd^+^ Venules**							
- Absolute number	14 (3–34)	0.0 (0.0–0.8)	*0.001*	19 (7–35)	17 (5–32)	0.39	0.001
- Number per mm^2^ (density)	6.0 (1.4–11.8)	0.0 (0.0–0.17)	*0.001*	8.3 (3.4–14.0)	7.7 (2.8–14.3)	0.85	0.001
- As % of ERG^+^ venules (proportion)	4.3 (1.0–8.6)	0.0 (0.0–0.08)	*0.001*	5.3 (2.2–8.3)	4.0 (1.3–7.8)	0.04	0.001
**Absolute number of intrafollicular PNAd^+^ Venules**	1 (0–3)	0.0 (0.0–0.0)	*0.001*	2 (0–5)	0 (0–3)	0.003	0.001
**Extrafollicular MAdCAM-1^+^ Venules**							
- Absolute number	20 (8.5–42.0)	18 (9.0–32.0)	*0.38*	18 (8–35)	32.0 (16.0–52.0)	0.001	0.001
- Number per mm^2^ (density)	10.7 (3.6–16.3)	10.0 (5.1–15.1)	*0.71*	9.0 (3.8–15.5)	14.2 (8.4–22.2)	0.001	0.001
- As % of ERG^+^ venules (proportion)	6.9 (2.6–11.0)	7.8 (4.5–11.8)	*0.15*	5.5 (2.5–9.5)	7.8 (4.5–12.0)	0.001	0.8
**Absolute number of intrafollicular MAdCAM-1^+^ Venules**	0.0 (0.0–2.0)	0.0 (0.0–1.0)	*0.02*	0 (0–3)	0.0 (0.0–3.0)	0.37	0.002
**Extrafollicular MAdCAM-1^+^ PNAd^+^ Venules**							
- Absolute number	5 (0–11)	0 (0–0)	*0.001*	6 (1–15)	8 (2–17)	0.04	0.001
- Number per mm^2^ (density)	2.03 (0.0–4.52)	0 (0–0)	*0.001*	2.5 (0.7–6.4)	3.6 (0.8–7.8)	0.002	0.001
- As % of ERG^+^ venules (proportion)	1.4 (0.0–2.7)	0 (0–0)	*0.001*	1.8 (0.4–3.8)	2.0 (0.5–3.9)	0.21	0.001
**Percentage MAdCAM-1^+^ of PNAd^+^ venules**	48.0 (13.9–69.4)	*	*-*	42.9 (19.1–70.5)	57.9 (35.7–83.3)	0.001	-

Values expressed in n (%) or as median (interquartile range); ERG, ETS related gene; HC, Healthy control; MAdCAM-1, mucosal vascular addressin cell adhesion molecule-1; PNAd, peripheral node addressin; UC, ulcerative colitis. * There are no PNAd^+^ venules in non-inflamed biopsies, therefore no percentage is presented.

**Table 4 cells-09-00891-t004:** Univariable and multivariable analysis of potential variables correlating with the expression of PNAd^+^ (Table 4A) and MAdCAM-1^+^ (Table 4B) venules as a proportion of ERG^+^ venules at diagnosis.

**A. *Baseline PNAd***	**Univariable**			**Multivariable**	
	**R^2^ (%)**	**β (SE)**	***p*-Value**	**β (SE)**	***p*-Value**
**Age at diagnosis**	0.7	−0.08 (0.001)	0.11		NS
**Sex***	2.6	−0.16 (0.006)	0.002		NS
**Smoking behavior****	1.8	−0.13 (0.004)	0.01		NS
**Symptom duration prior to initial diagnosis in weeks**	6.1	0.25 (0.001)	0.001	0.22 (0.001)	0.01
**Disease location º**	12.4	−0.35 (0.004)	0.001	−0.40 (0.005)	0.001
**Histologic disease activity**					
**- Geboes 0**	3.9	0.20 (0.004)	0.002	0.21 (0.005)	0.01
**- Geboes 1**	4.7	0.22 (0.006)	0.001		NS
**- Geboes 2A**	1	0.10 (0.005)	0.11		NS
**- Geboes 2B**	0.2	0.05 (0.004)	0.44	-	
**- Geboes 3**	0	0.02 (0.005)	0.73	-	
**- Geboes 4**	1.1	0.11 (0.008)	0.1		NS
**- Geboes 5**	1.6	0.13 (0.002)	0.05		NS
**First effective remission induction treatmentºº**	1.4	-0.12 (0.002)	0.02		NS
**Number exacerbations per year clinical follow-up**	0.8	0.09 (0.015)	0.08		NS
*Female = 0 , male = 1 **Never = 0, ceased = 1, yes = 2 ºProctitis = 1, left-sided = 2, extensive = 3, ººNo treatment = 0, 5ASA = 1, topical steroids = 2, oral steroids = 3, anti-TNF = 4, anti-integrins = 5, surgery = 6; β = Regression coefficient, SE = Standard Error.					
**B. Baseline MAdCAM-1**	**Univariable**			**Multivariable**	
	**R^2^ (%)**	**β (SE)**	***p*-value**	**β (SE)**	***p*-value**
**Age at Diagnosis**	0.2	0.04 (0.001)	0.46		
**Sex***	0.3	−0.05 (0.006)	0.34	-	-
**Smoking Behavior****	0.3	0.05 (0.004)	0.35	-	-
**Symptom Duration Prior to Initial Diagnosis in Weeks**	0.3	0.06 (0.001)	0.44	-	-
**Disease Location º**	0	−0.01 (0.004)	0.84	-	-
**Histologic Disease Activity**					
**- Geboes 0**	1.3	−0.11 (0.004)	0.07		NS
**- Geboes 1**	1.3	−0.11 (0.006)	0.08		NS
**- Geboes 2A**	3.1	−0.18 (0.005)	0.01		NS
**- Geboes 2B**	2.4	−0.15 (0.004)	0.02		NS
**- Geboes 3**	3.4	−0.19 (0.005)	0.003		NS
**- Geboes 4**	1.2	−0.11 (0.005)	0.08		NS
**- Geboes 5**	1.3	−0.11 (0.002)	0.08		NS
**First Effective Remission Induction Treatment ºº**	0.3	−0.06 (0.002)	0.29	-	-
**Number Exacerbations Per Year Clinical Follow-Up**	0.6	0.08 (0.187)	0.14		NS
*Female = 0, male = 1 **Never = 0, ceased = 1, yes = 2, ºProctitis = 1, left-sided = 2, extensive = 3, ººNo treatment = 0, 5ASA = 1, topical steroids = 2, oral steroids = 3, anti-TNF = 4, anti-integrins = 5, surgery = 6; β = Regressioncoefficient, SE = Standard Error.					

**Table 5 cells-09-00891-t005:** Values of CXCL-13 and CCL-19 in the serum and supernatant of patient with ulcerative colitis at first presentation, compared with healthy controls (in pg/ml).

	Inflamed UC *n* = 22	Healthy Controls *n* = 10	*p*-Value
**Serum in pg/mL**			
**CXCL-13**	86.2 (58.3–109.4)	36.9 (31.3–48.8)	0.001
**CCL-19**	167.7 (133.8–204.3)	100.3 (37.4–158.1)	0.02
**Supernatant Biopsy in pg/mL**			
**CXCL-13**	4.3 (0.6–42.72)	0.6 (0.6–20.8)	0.363
**CCL-19**	16.4 (13.4–17.2)	10.5 (9.4–12.9)	0.001

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
