# Peer review of "Development of Mucosal PNAd+ and MAdCAM-1+ Venules during Disease Course in Ulcerative Colitis"

_cells, 2020, doi:10.3390/cells9040891_

Round 1

Reviewer 1 Report

The authors demonstrated that mucosal PNAd+ HEVs was associated with the severity of colitis in UC and a high expression of MAdCAM-1+ venules was predictive for the exacerbations of colitis and increases with each subsequent exacerbation. The authors also showed that the expressions of PNAd+ HEVs were associated with the formation of colonic follicles in the inflamed intestinal mucosa. This is very interesting result, however the correlation between endoscopic severity/histological inflammation and the expression of PNAd+/MAdCAM-1 was unclear. I raise several concerns including this point as listed below.

  1. In Table1, there were only two patients in Geboes scores of 3.1 or less. How did the authors analyze the correlation between PNAd + and MAdCAM-1 expression and histological inflammation of Geboes score 0-3?

  1. The authors need to show the correlation between endoscopic severity (UCEIS or Mayo score) at the site of biopsy and the expression of PNAd+/MAdCAM-1.

  1. PNAd+/MAdCAM-1 expression in intestinal mucosa has been only proven by immunostaining. The authors need to prove mRNA expressions of them by RT-PCR.

  1. CXCL-13 is known as B cell chemoattractant. Why did the authors measure CXCL-13? Have the authors measured the production of CCL21 as known to T cell chemoattractant?

  1. The authors mentioned that the higher expression of MAdCAM-1 was predictive for the exacerbations of colitis. However, the authors selected one biopsy sample from each patients. So, this result did not reflect the entire intestinal inflammation in UC. I think that the authors need to clarify the correlation between the exacerbations rate and the average expression of MAdCAM-1 in multi-samples of inflamed mucosa.

Reviewer 2 Report

The authors presented the importance of the mucosal PNAd+ and MAdCAM-1+ venules in the disease activity and severity in the UC.

Overall, the manuscript is well written in terms of the results and logical flow. The results obtained in this study may contribute to the development of the disease marker using PNAd+ venules and chronicity using MAdCAM-1+ venules, therefore, can be accepted in its current form for publishing in the journal.

Author Response

We thank the reviewer for the thoughtful review of our manuscript.

Reviewer 3 Report

In the present study the AA analyze in endoscopic biopsies from  a large number of UC patients at diagnosis vs HC, the % of mucosal PNAd+ and MAdCAM-1+venules and correlate the findings with clinical data at diagnosis. Furthermore they also evaluate subsequent biopsies  obtained from the same patients during follow-up. Finally they also analyze the course of disease according with the expression of PNAd+ and MAdCAM-1+ 2 venules at diagnosis. They found that PNAd+HEVs expression is mainly  associated with active disease, while MAdCAM-1+venules stay after inflammation is resolved and increase after  subsequent flares thus reflecting chronicity. Moreover a higher proportion of PNAd+ venules in baseline biopsies seems to be associated with response to aminosalicylates and /or topical steroids vs oral steroids, anti-TNF-a and surgery need

The study is well designed, and properly analyzed and discussed 

Minor points:

1.      apparent discrepancies need to be reconciled:

a.       Results: Page 16 line250-254: patients with higher proportion of PNAd+ venules respond to aminosalicylates and/or topical steroids……

Discussion: page 20 line 251-362 , results described above need to be mentioned and discussed here.

b.      Results: Page 10 line219-222. Discussion page 19 line 299-300

2.      Supplementary Table S1: legend: incorrect mention  to MAdCAM-1, PNAd

3.      Results listed in the text  at “The presence of PNAd+ and MAdCAM-1+ venules in colonic biopsies at follow-up”  would benefit from a boxplot figure and “ Association of PNAd+ and MAdCAM-1+ venules with disease course” would  also benefit from a figure showing  the increasing number of MAdCAM-1+ venules associated with exacerbation(s)

Round 2

Reviewer 1 Report

The authors have addressed most of my comments.